New information on the anatomy of the Chinese Early Cretaceous Bohaiornithidae (Aves: Enantiornithes) from a subadult specimen of Zhouornis hani

Zhang Yuguang 1
O’Connor Jingmai 2
Di Liu 1
Qingjin Meng 1
Sigurdsen Trond 3
Chiappe Luis M. 3 chiappe@nhm.org
1 Beijing Natural History Museum , Beijing , PR China
2 Institute of Vertebrate Paleontology and Paleoanthroplogy , Beijing , PR China
3 Dinosaur Institute, Natural History Museum of Los Angeles County , Los Angeles, CA , USA
Thewissen J
Electronic publication date: 2014 May 29
Publication date: 2014
Volume: 2
Electronic Location ID: e407
Received 2014 Feb 6; Accepted 2014 May 9
Copyright: © 2014 Zhang et al.
Copyright year: 2014
Copyright holder: Zhang et al.
License: This is an open access article distributed under the terms of the Creative Commons Attribution License, which permits unrestricted use, distribution, reproduction and adaptation in any medium and for any purpose provided that it is properly attributed. For attribution, the original author(s), title, publication source (PeerJ) and either DOI or URL of the article must be cited.
License URL: https://creativecommons.org/licenses/by/4.0/

Keywords: Bohaiornithidae, Enantiornithes, Mesozoic birds, Pedal morphology, Skull, Jehol

Funding: Project IG 201101C2 Dinosaur Institute of the Natural History Museum Funding for this research was provided by Project IG 201101C2 (Study of the morphology and systematics of the Jehol Biota) of the Building of Innovative Team Plan Program to the Beijing Natural History Museum of Los Angeles County, and by donations from Doreen and Glenn Gee to the Dinosaur Institute of the Natural History Museum of Los Angeles County. The funders had no role in study design, data collection and analysis, decision to publish, or preparation of the manuscript.

==============================
Enantiornithines are the most diverse avian clade in the Cretaceous. However, morphological specializations indicative of specific ecological roles are not well known for this clade. Here we report on an exquisitely well-preserved specimen from the Lower Cretaceous Jehol Group of northeastern China, which pedal morphology is suggestive of a unique ecological specialization within Enantiornithes. The morphology of the new specimen is largely indistinguishable from that of the holotype of the bohaiornithid enantiornithine Zhouornis hani, albeit the latter is somewhat larger. The new specimen provides important and previously unknown details of the skull of Zhouornis hani, which add to the limited knowledge about the cranial anatomy and evolution of enantiornithines. The information offered by the new specimen also augments our understanding of the postcranial morphology of bohaiornithid enantiornithines, a clade that has been only recently recognized. With the description of this specimen, Zhouornis hani becomes one of the most anatomically complete known enantiornithine species, which will facilitate future morphological studies.

Introduction

During the last two decades, over 40 species of birds have been named from the Lower Cretaceous Jehol Group Lagerstatten of northeastern China, and discoveries continue at an unprecedented rate (Zhou & Zhang, 2006; Chiappe, 2007; O’Connor, Chiappe & Bell, 2011). The most diverse recognized clade is Enantiornithes, inferred to be the dominant clade of Cretaceous birds (Chiappe, 2007; O’Connor et al., 2009). The Jehol Group, however, is not only important because it has produced such a wealth of fossil evidence, but because these fossils reveal critical information regarding aspects of the integument, diet, ecology, reproduction, and ontogeny (Zhang & Zhou, 2000; Zhou & Zhang, 2006; O’Connor et al., 2009; Zheng et al., 2011; Zheng et al., 2012; Zheng et al., 2013a; Zheng et al., 2013b).

Here we describe an exquisite new specimen (BMNHC Ph 756), which is morphologically indistinguishable from that of the Jehol enantiornithine Zhouornis hani (Zhang et al., 2013). This taxon has recently been assigned to Bohaiornithidae; with six included genera (Bohaiornis, Longusunguis, Parabohaiornis, Shenqiornis, Sulcavis, and Zhouornis) (Hu et al., 2011; Li et al., 2014; O’Connor et al., 2013; Wang et al., 2010); it represents the most speciose recognized clade of enantiornithines to receive phylogenetic support (Wang et al., 2014). By preserving previously unknown morphological details, in particular of the skull, BMNHC Ph 756 provides valuable information for understanding the cranial anatomy and evolution of bohaiornithids, and enantiornithines and basal birds in general. Referred specimen BMNHC Ph 756 is preserved in ventral view, perfectly complimenting the holotype of Zhouornis hani which is preserved in dorsal view (Zhang et al., 2013), thus with this description this taxon becomes one of the most anatomically well-known Early Cretaceous birds.

Aves Linnaeus, 1758	
Ornithothoraces Chiappe, 1995	
Enantiornithes Walker, 1981	
Bohaiornithidae Wang et al., 2014	
Zhouornis hani Zhang et al., 2013	

Materials and Methods

Taphonomy and preservation

BMNHC Ph 756 consists of a nearly complete sub-adult individual preserved fully articulated in ventral view in a single slab (Figs. 1 and 2) collected from Jehol deposits near Xiaoyugou (Chaoyang, Liaoning Province, China). Based on exposures around the Xiaoyugou locality, BMNHC Ph 756 is considered to come from the Jiufotang Formation. BMNHC Ph 756 preserves faint remnants of plumage and it is approximately 80% the size of the holotype of Zhouornis hani, CNUVB-0903 (Zhang et al., 2013) (Fig. 3). BMNHC Ph 756 was mechanically prepared at the Natural History Museum of Los Angeles County (Table 1) and an epoxy resin cast is housed at this museum (LACM 7892/155842). Preparation revealed no evidence of tampering and asserted the legitimacy of the fossil.

Figure 1 Photograph of BMNHC Ph 756.

Abbreviations: ce, cervical vertebrae; co, coracoid; fem, femur; fi, fibula; fu, furcula; h, humerus; il, ilium; pu, pubis; py, pygostyle; r, radius; sc, scapula; st, sternum; ti, tibiotarsus; tmt, tarsometatarsus; u, ulna; I–IV, digits I–IV.

Figure 2 X-ray photograph of BMNHC Ph 756.

Figure 3 Photograph of the holotype of Zhouornis hani, CNUVB-0903.

Abbreviations: co, coracoid; fem, femur; fi, fibula; fu, furcula; h, humerus; pu, pubis; py, pygostyle; r, radius; sc, scapula; st, sternum; ti, tibiotarsus; tmt, tarsometatarsus; u, ulna; I–IV, digits I–IV.

Table 1 Comparative measurements (in millimeters) of BMNHC Ph 756 and CNUVB-0903 (holotype).

	BNHM Ph756	CNUVB-0903	
Pygostyle	22.4	17.3	
Scapula	32.4(l) 36.0(r)	40.4 (l) 40.7 (r)	
Coracoid	20.9(l) 23.1(r)	28.3 (l) 28.4 (r)	
Humerus	38.3(l) 40.5(r)	50.6 (l) 50.6 (r)	
Ulna	36.9(l) 41.5(r)	54.0 (l) 53.5 (r)	
Carpometacarpus	19.3(l) 19.5(r)	24.4 (l) 23.6 (r)	
Femur	31.4(l) 26.4(r)	43.5 (l) 44.5 (r)	
Tibiotarsus	39.3(l) 41.7(r)	51.2 (l) 52.1 (r)	
Tarsometatarsus	22.5(l) 20.0(r)	25.8 (l) 26.1 (r)	
Notes.

l left

r right

Results

Anatomical description

The anatomical nomenclature used here primarily follows that of the Nomina Anatomica Avium (Baumel & Witmer, 1993).

Skull

The skull is preserved in right lateral view (Fig. 4); the bones are only slightly disarticulated, more so in the caudal region of the skull. The well-preserved premaxilla of BMNHC Ph 756 confirms the presence of four teeth in Zhouornis hani, which was unclear in the holotype (Zhang et al., 2013). This bone is rostrally restricted on the facial margin as in other bohaiornithids, although the palatal process is visible extending caudally for about half the length of the rostrum (leveled with the rostral margin of the antorbital fossa). The lateral margin of the corpus bears several small nutrient foramina. The frontal processes of the premaxillae are unfused along their medial lengths, although whether the corpus was fused is unclear. The caudal extension of those processes is unclear due to overlapping by the nasals. The nasals are long and clearly joined to one another, a condition not visible in the holotype specimen or any other bohaiornithid (Wang et al., 2014). Although these bones are displaced, the long premaxillary process indicates that the nasal lined most of the dorsal margin of the external nares. The nasal expands laterally to form the caudal margin of the external nares but it is unclear if a maxillary process was present (variably present among known bohaiornithids; Wang et al., 2014). The nasal reaches its maximum width at its midpoint and, despite not being in their natural position, it is clear these bones formed part of the dorsal margin of the antorbital fossa. These morphologies are consistent with what is visible in the holotype (Zhang et al., 2013).

Figure 4 Detail of skull of BMNHC Ph 756.

Abbreviations: af, antorbital fenestra; bp, basipterygoid process; d, dentary; en, external nares; f, frontal; hy, hyoid; j, jugal; la, lacrimal; mg, Meckel’s groove; mx, maxilla; n, nasals; or, orbit; p, parietal; pm, premaxilla; q, quadrate; sp, splenial; su, surangular.

The articulation between the premaxilla and maxilla is broken; the premaxillary ramus of the maxilla, if it were complete, is interpreted as shorter than the jugal ramus. The ascending nasal process forms much of the caudal margin of the external nares, as in Shenqiornis (Wang et al., 2010). Whether the caudal portion of this process was recessed and perforated by accessory antorbital fenestrae, as in the holotype, is unclear. The maxilla forms the rostral and ventral margins of the antorbital fossa, the caudal margin of which is formed by the lachrymal. The antorbital fossa is medially lined by thin sheets of bone, presumably displaced scleral ossicles and other indeterminate bones. There are three teeth preserved in the right maxilla, a number comparable to that in the holotype and other bohaiornithids. The caudalmost tooth is leveled with the rostral base of the nasal (dorsal) process. The jugal ramus is marked by numerous small foramina; the caudal margin of the maxilla tapers caudoventrally, forming a sloping articulation with the rostrodorsally tapering proximal end of the jugal, as in other bohaiornithids (Wang et al., 2014). The descending process of the lachrymal articulates with both the jugal and maxilla. BMNHC Ph 756 thus confirms the relations between the maxilla, lachrymal, and jugal, and it also provides details of the articulations of these bones to one another, otherwise unclear in all other known bohaiornithids (Wang et al., 2014).

The lachrymal (lacrimal) is T-shaped, as in the holotype specimen, with the dorsal rami strongly angled, caudodorsal-rostroventrally. The caudodorsal ramus is tapered sharply distally; the rostrodorsal ramus is poorly preserved—in the holotype specimen, it is longer than the caudodorsal ramus. In BMNHC Ph 756 the rostrodorsal ramus is mediolaterally compressed while the caudodorsal margin is dorsoventrally compressed, as in Parabohaiornis and Longusunguis (Wang et al., 2014). Like in Pengornis and other bohaiornithids, the descending ramus of the lachrymal is expanded laterally relative to the rostrodorsal process, continuous with the caudodorsal ramus so that the descending ramus has an L-shaped cross-section (O’Connor & Chiappe, 2011; Wang et al., 2014). The descending ramus is also expanded distally where it contacts the jugal and maxilla. The jugal is slightly deflected dorsally where it articulates with the maxilla. The bone is straight and thinner than the jugal ramus of the maxilla; the distal end is covered by the quadrate.

The frontals are completely unfused and disarticulated, both preserved in dorsal view. No postorbital bone is preserved although it is possible one was present and covered by other skull bones—a large postorbital is preserved in bohaiornithids Shenqiornis and Sulcavis (Wang et al., 2010; O’Connor et al., 2013). Like other enantiornithines, the frontals are petal shaped, rostrally narrow and caudally expanded to more than twice the rostral mediolateral width. The lateral margin is strongly concave; the edge forms a small ridge also present in Pengornis (O’Connor & Chiappe, 2011) and Parabohaiornis (Wang et al., 2014). The medial margin is straight, where the two frontals contact; the caudal margin is rounded. Both parietals are preserved, disarticulated and overlapping with other elements. One parietal is preserved, partially covered by the left frontal; the one corner visible forms a sharp 90° angle. The other overlies the distal ends of the post-dentary bones, possibly covering part of the posterior portion of the skull; the parietal appears to be trapezoidal, although the two parallel sides do not differ greatly in length.

A few fragments in the orbit are interpreted as scleral ossicles. Both quadrates are preserved. The right quadrate is largely covered by other bones, although clearly visible in the X-ray (Fig. 2); the left one is well exposed and in medial view. A foramen is visible perforating the medial surface of the left quadrate, proximal to the mandibular condyles. This foramen is interpreted as a pneumatopore, indicating that the quadrate was pneumatized by a diverticulum as in some enantiornithines including bohaiornithid Shenqiornis and modern birds (Wang et al., 2010; O’Connor & Chiappe, 2011).

The dentary bones may have been fused rostrally but a crack obfuscates interpretation (Fig. 4). The right dentary is in lateral view while the left is in medial view, revealing Meckel’s groove and the caudally restricted splenial. The right dentary preserves six teeth and the left one preserves seven; six teeth are preserved in the holotype (Zhang et al., 2013). Given that the rostral portions of the dentaries are missing, the total number of teeth may have been slightly greater, however, this number is consistent with other bohaiornithids (Wang et al., 2014). The rostral half of the dentary has parallel dorsal and ventral margins and bears numerous small lateral foramina; the caudal half of the dentary is deflected ventrally so that overall the ventral margin is somewhat concave, as in most enantiornithines (O’Connor & Chiappe, 2011). The well-preserved medial, dentigerous surface of the left dentary does not show any evidence of the individualized interdental plates reported for Archaeopteryx lithographica (Elzanowski, 2002). The dentary tapers caudoventrally, forming a sloping, unforked articulation with the rostrodorsally tapering rostral end of the surangular as in most enantiornithines (O’Connor & Chiappe, 2011). The surangular appears to be approximately two-thirds the length of the dentary. The precise contact between the surangular and the angular are not entirely clear but it seems as if the latter also contacts the dentary rostrally. The left mandible shows a triangular-shaped splenial, perforated by a small foramen near its center. The caudal ramus of the splenial ends with the dentary and these bones form a long caudomedial suture with a long and caudoventrally slanted prearticular.

The teeth of BMNHC Ph 756 are exquisitely preserved (Fig. 4). The first two premaxillary teeth are more slender than other upper jaw teeth, as in other bohaiornithids (Wang et al., 2014). The remaining teeth in the upper jaw are fairly robust and similar to those observed in Shenqiornis and other bohaiornithids (Wang et al., 2010; Wang et al., 2014). Their crowns are fat proximally, tapering rapidly at their slightly deflected occlusal ends; the mandibular crowns taper more gradually, as in Parabohaiornis (Wang et al., 2014). All teeth have D-shaped cross-sections, with the flat side forming the lingual margin. However, like other known birds, no striations, like those in Sulcavis, are present (O’Connor et al., 2013).

Vertebral column

The proximal-most cervicals (atlas and axis) are not visible due to the disarticulation of the skull roof; six to seven vertebrae are visible between the skull and the furcula but it is not possible to determine with certainty the boundary between the cervical and thoracic series due to poor preservation (Fig. 1). The cervicals are in ventral view. These vertebrae are only slightly longer than wide; although these proportions may be the result of postmortem dorsoventral compression, similarly robust cervicals are observed in bohaiornithids Longusunguis and Shenqiornis (Wang et al., 2010; Wang et al., 2014). The ventral surface of the centrum is clearly keeled as in some other enantiornithines (e.g., Pengornis, Eoalulavis). Long, tapering costal processes are present, approximately 1/2 to 2/3 the length of the centrum; it is unclear if they are fused to the vertebrae. One thoracic vertebra is visible through the thin sternal bone that overlies it (Fig. 5); it is clearly spool-shaped, although no other information is visible. Distally, two poorly preserved thoracic vertebrae are preserved in articulation with the synsacrum. The proximal most preserved of these vertebrae is elongate.

Figure 5 Detail photograph of pelvic girdle of BMNHC Ph 756.

Abbreviations: fem, femur; il, ilium; isc, ischium; pu, pubis; pvp, ventral processes of pygostyle; py, pygostyle; sy, synsacrum; up, uncinate process.

The synsacrum, poorly preserved in the holotype, is formed of seven vertebrae, which decrease in length distally (Fig. 6). The ventral surfaces of the synsacral vertebrae are excavated by a shallow, continuous groove. The transverse processes increase in lateral extent distally, peaking at the sixth vertebrae, where the robust processes are oriented perpendicular to the synsacrum and would have contacted the ilia, also visible in the referred specimen of Parabohaiornis IVPP V18690 (Wang et al., 2014). The transverse processes of the last synsacral vertebra are oriented caudolaterally. There are four to five free caudals preserved, although only the proximal three preserve any morphology. The articular surfaces are flat to concave. The transverse processes decrease in length distally; in the proximal caudals the transverse processes are approximately 150% the width of the centrum and are caudolaterally oriented.

Figure 6 Detail photograph of pelvic girdle of BMNHC Ph 756.

Abbreviations: fem, femur; il, ilium; isc, ischium; pu, pubis; pvp, ventral processes of pygostyle; py, pygostyle; sy, synsacrum; up, uncinate process.

The pygostyle is preserved in right ventrolateral view. Like other enantiornithines, it has forked proximal dorsal processes (only one half of the fork is exposed) and ventrolateral processes (so that the ventral margin is excavated and deeply concave for almost its entire length). However, unlike most other enantiornithines that possess a distinct distal constriction in the pygostyle (e.g., Rapaxavis, Longipteryx, Halimornis), the element appears to taper smoothly distally, similar to the holotype and Vescornis; this feature appears to be a synapomorphy of Bohaiornithidae (Wang et al., 2014). In the distal third of the pygostyle, the neural spines of two verterbrae can be differentiated by sutures, consistent with interpretations of this specimen as a subadult.

Individual elements of the gastralial system are preserved overlapping the sacral–caudal transition. A single possible uncinate process, like that in Longipteryx (Zhang et al., 2001), is preserved on the left side, level with the thoracic-sacral contact (Fig. 6). Uncinate processes are clearly preserved in articulation in the holotype of Parabohaiornis (Wang et al., 2014).

Pectoral girdle and sternum

Like in the holotype specimen, the furcula is Y-shaped, with an elongate hypocleidium that is approximately half the length of the furcular rami (Fig. 5). The rami expand towards their omal tips so that the cranial margin is close to twice as wide omally than it is at the clavicular symphysis. The furcular rami, although in cranial view, are convex and interpreted as caudally excavated as in other enantiornithines (Chiappe & Walker, 2002). The hypocleidium bears a small ridge along the length of its cranial surface; a similar ridge is observed in the bohaiornithid enantiornithines Shenqiornis mengi (Wang et al., 2010) and Parabohaiornis (Wang et al., 2014).

The scapulae are partially covered by other bones; their morphology is more visible in the X ray (Fig. 2). The coracoidal articular facet is exposed and flat. As in the holotype, the acromion is long and it does not appear costolaterally wide, although the distal tip is not visible. In the X ray (Fig. 2), the left scapular blade appears slightly curved and tapered, similar to that of the holotype (Zhang et al., 2013), and unlike the straight scapular blade of most enantiornithines (Chiappe & Walker, 2002). This feature is considered a synapomorphy of Bohaiornithidae (Wang et al., 2014).

The coracoids are both in articulation with the sternum in ventral view, as opposed to those in the holotype specimen, in which they are exposed in dorsal view. As in most other enantiornithines (Chiappe & Walker, 2002), no procoracoid process is present. The narrow neck of the coracoid extends for slightly more than the proximal half of the total length before the corpus rapidly expands medially and laterally so that both margins are generally concave (Fig. 5). Whether the sternal corner of the lateral margin of the coracoid of BMNHC Ph 756 was more convex, as in the holotype and other bohaiornithid enantiornithines, cannot be ascertained because this region is either poorly preserved or overlapped by other elements. The sternal margin is slightly concave and the two coracoids articulate adjacent to each other, although it is possible the medial angles slightly overlapped.

The sternum is similar to that of the holotype specimen in all preserved aspects (Zhang et al., 2013). The rostral margin is rounded, demarcating caudally an angle of approximately 110° and the lateral caudal trabeculae are slightly splayed caudolaterally, as in other bohaiornithid enantiornithines (Wang et al., 2014). BMNHC Ph 756 shows that the lateral margin of the sternum of Zhouornis hani was not expanded into a lateral (zyphoid; Clarke, Zhou & Zhang, 2006) process, and that the intermediate trabeculae are small triangular expansions located on the caudal margin, midway between the lateral trabecula and the xiphoid process, as in most other enantiornithines. Distally, the latter two processes extend to the same level; the trabeculae are shorter in bohaiornithids Parabohaiornis and Longusunguis (Wang et al., 2014). The xiphoid process is long and straight and bears no distal expansion. The sternal carina is limited to the caudal half of the sternum; it is unclear whether it was forked proximally or not.

Forelimb

The humeri are both in cranial view (Fig. 7), as opposed to those in the holotype that are exposed caudally (Zhang et al., 2013). The proximal cranial surface is slightly concave but no transverse ligamental groove or circular fossa like that in some enantiornithines and other basal birds (Chiappe, 1996) is visible. The profile of the humeral head is typically enantiornithine, concave on the midline and rising dorsally and ventrally (Chiappe, 1996; Chiappe & Walker, 2002). The deltopectoral crest, projected dorsally, is slightly less than the width of the shaft as in other bohaiornithids. Distally, the condyles are on the cranial margin. The distal margin is slightly angled, as in Parabohaiornis IVPP V18690, although the flexor process is not distinctly enlarged as in some Late Cretaceous enantiornithines (e.g., Alexornis, Martinavis). The dorsal condyle is round; the ventral condyles on both the right and left humerus have been lost to abrasion. The morphology of the bones of the forearm and hand are similar to those of the holotype. The ulna is slightly bowed proximally and it displays a minimal development of the olecranon process. The straight radius displays a longitudinal groove on its interosseum surface present in some enantiornithines including Bohaiornis, Parabohaiornis, and Longusunguis (Wang et al., 2014).

Figure 7 Detail photograph of the right (A) and left (B) forelimb of BMNHC Ph 756.

Abbreviations: bt, bicipital tubercle; cd, dorsal condyle; cm, medial condyle; dc, deltopectoral crest; fp, flexor process; h, humerus; mc I–III, metacarpals I–III; r, radius; ra, radiale; rg, radial groove; sl, semilunate carpal; u, ulna; ul, ulnare; I–III, digits I–III.

Two subequal and subtriangular proximal carpal bones are preserved (Fig. 7). The semilunate metacarpal contacts the major and minor metacarpals but unlike the condition in the holotype, it remains unfused. Such a difference is likely ontogenetic, given that BMNHC Ph 756 is approximately 80% the size of the holotype (based on the most complete long bones). The semilunate is D-shaped in profile, but slightly asymmetrical, tapering slightly caudally. The alular metacarpal is trapezoidal so that the caudal margin is longer than the cranial margin; the latter margin is slightly more rounded than in the holotype. Comparable to the holotype and other bohaiornithids, the alular digit is short, not extending beyond the distal end of the major metacarpal, the proximal phalanx is slightly bowed and the claw is highly recurved (Zhang et al., 2013; Wang et al., 2014).

The major metacarpal is straight and robust; the minor metacarpal also appears robust and bowed proximally (Fig. 7). Differences in the morphology on the right and left, however, suggest that the right minor metacarpal is in caudoventral view, causing the unusual thickness. The bowed shape of the minor metacarpal demarcates a thin intermetacarpal space. No intermetacarpal process is visible. Distally, the metacarpals are unfused and the minor metacarpal projects significantly beyond the distal end of the major metacarpal, as in all other enantiornithines (Chiappe & Walker, 2002). The major digit has three phalanges; the proximal phalanx is the longest and most robust. The caudal surface of the phalanx is thickened, especially along the proximal half. This condition is visible on both sides of BMNHC Ph 756 and it is also present in the holotype, thus we regard it as an autapomorphy of the Zhouornis hani. The intermediate phalanx is approximately 2/3 the length of the proximal phalanx and articulates with a slightly smaller and less recurved claw than that of the alular digit, as in other bohaiornithids. As in the holotype and all other bohaiornithid specimens, the minor digit only preserves a single phalanx on either side, however, the well-developed distal articular surface of this delicate phalanx suggest a second phalanx, albeit reduced, might have been present.

Pelvic girdle

BMNHC Ph 756 provides substantial new information on the structure of the pelvis of Zhouornis hani (Fig. 6) and Bohaornithidae. The pelvic elements are disarticulated indicating they were neither fused at the level of the acetabulum nor to the synsacrum, at the time of death. The ilium has a straight dorsal margin, slightly concave over the acetabulum. This morphology can also be identified in the poorly preserved ilium in the referred specimen of Parabohaiornis IVPP V18690 (Wang et al., 2014) although it was apparently absent in Bohaiornis (Li et al., 2014). In lateral view, the preacetabular wing is approximately twice as dorsoventrally tall as the postacetabular wing, with a convex cranial margin (Fig. 6). The postacetabular wing is shorter and triangular with a blunt caudal margin; its ventral edge is slightly concave. Other bones largely cover the ischia and thus only minimal anatomical information is available. What is preserved of the ischium is two-thirds the length of the pubes. Its medial surface appears to bear a longitudinal ridge but a proximodorsal process is not visible although the latter feature is present in other bohaiornithids (e.g., Bohaiornis, Shenqiornis, Parabohaiornis). Approximately mid-length the shaft of the ischium appears to expand medially. The pubes are long, contacting distally, but it cannot be determined if they were expanded into a cranial boot as in the holotype (Zhang et al., 2013). The pubic apron is short, approximately 15–20% the length of the bone, a ratio comparable to that of the holotype. The shaft has a suboval to rounded cross section.

Hind limb

Both hind limbs are preserved in their entirety (Fig. 1). The femora are robust, straight and about 3/4 the length of the tibiotarsus. The left tibiotarsus is better preserved, exposing the craniomedial surface. No cnemial crests are present, typical of Early Cretaceous enantiornithines. Laterally, the fibular crest extends for less than a quarter the length of the tibiotarsus. The crest is well developed, generally wider than in other enantiornithines but comparable to that preserved in the referred specimen of Parabohaiornis IVPP V18690 (Wang et al., 2014). Distally the right tibiotarsus shows that the proximal tarsals are not fully fused to the tibia (Fig. 8B). The triangular ascending process is not well preserved on either side; the cranial surface of the distal right tibia is excavated by a fossa. The distal condyles (best visible on the right element) are poorly preserved, but the medial one appears larger than the lateral condyle, as in Bohaiornis, Parabohaiornis, and Longusunguis (Hu et al., 2011; Wang et al., 2014), as well as other basal birds. The fibulae are poorly preserved and incomplete.

Figure 8 Detail photograph of the right (A) and left (B) feet of BMNHC Ph 756.

Abbreviations: mt I–IV, metatarsals I–IV; tc, possible tubercle for the attachment of m. tibialis cranialis; I–IV, digits I–IV.

No distal tarsals are preserved capping the metatarsals. Like in the holotype, metatarsals II–IV are coplanar and well demarcated by straight sutures throughout their lengths (Fig. 8). Metatarsal III is the longest and most robust; metatarsals II and IV are subequal in length but the latter is more slender, as in the holotype and other enantiornithines (Chiappe & Walker, 2002). In spite of the good preservation of the feet, the attachment of the m. tibialis cranialis is not clear in either BMNHC Ph 756 or the holotype (Zhang et al., 2013), indicating this feature was poorly developed in Zhouornis hani relative to other enantiornithines (Chiappe, 1993). A slight bulge on the craniomedial margin of the right metatarsal II obscured by breakage may represent the attachment for this muscle or it may represent part of the medial plantar crest. This tubercle is only preserved in a single other bohaiornithid, the referred specimen of Parabohaiornis IVPP V18690, in which it is located a third of the way down on the dorsolateral surface of metatarsal II, contacting metatarsal III (Wang et al., 2014). As in the holotype, bohaiornithids, and many other enantiornithines (Chiappe, 1993), the trochlea of metatarsal II is the widest. Metatarsal I articulates on the medial surface of metatarsal II (Fig. 8A). The metatarsal itself is short, robust, p-shaped.

Digit I appears to be located somewhat lower than in the holotype; the trochlea of metatarsal I ends slightly more proximal than that of metatarsal II. Digit I is short with a large claw that is slightly more recurved that the other pedal claws (Fig. 8). As in other bohaiornithids, digit II is the most robust in the foot, with the proximal phalanx large and proximally expanded. The intermediate phalanx is slightly longer but less robust and the claw is the largest in the foot, although not as long as that of digit III when the horny sheaths are considered. Digit III is the longest digit. The proximal phalanx is the longest in the foot, and the following two phalanges are subequal in length to the proximal phalanx of digit II. When the horny sheaths are considered, the claw of digit III is longer than the other pedal claws, but comparatively fairly unrecurved; as in other bohaiornithids, this ungual is more than half the length of the tarsometatarsus (Wang et al., 2014). Digit IV is poorly preserved or covered on both sides, however, the phalanges and ungual are all shorter and more delicate than any of the phalanges from the other digits (Fig. 8).

Discussion

BMNHC Ph 756 preserves numerous enantiornithine synapomorphies, including a dorsally forked pygostyle with a pair of ventrolateral processes, Y-shaped furcula, caudally restricted sternal keel, distally expanded lateral trabecular of the sternum, interosseal groove on radius, minor metacarpal projecting distally farther than the major metacarpal, and a reduced metatarsal IV. Smaller size notwithstanding, it is not possible to differentiate the specimen from the holotype of Zhouornis hani. BMNHC Ph 756 reveals details of the skull and postcranium that have remained unclear for Zhouornis hani, and for previously known enantiornithine specimens (O’Connor & Chiappe, 2011), such as the articulation between the jugal, the lachrymal, and maxilla, the number of premaxillary teeth and details of the dental morphology, the pneumatic condition of the quadrate, the number of synsacral vertebrae, and aspects of the sternum and the pelvis.

The lack of fusion in several of the compound bones (e.g., carpometacarpus, tibiotarsus, tarsometatarsus, pygostyle) supports the interpretation of BMNHC Ph 756 as a subadult, in which skeletal maturation was not completed at the time of death. Whereas the bone microstructure of the holotype was studied by Zhang et al. (2013), BMNHC Ph 756 was unavailable for histological sectioning. Therefore, presently we cannot contrast our ontogenetic interpretation of BMNHC Ph 756 with histological evidence. Given the absence of other information about the life history of Zhouornis hani, we tentatively interpret the smaller size and skeletal immaturity of BMNHC Ph 756 as indicative of a younger age relative to the holotype.

Figure 9 Interrelationships among enantiornithines, and between higher groups of Mesozoic birds, according to Wang et al. (2014).

Zhouornis has been recently assigned to the diverse enantiornithine clade, the Bohaiornithidae, with the support of a large cladistic analysis (Wang et al., 2014) (Fig. 9). Bohaiornithids are united by common morphologies of the rostrum (e.g., robust), teeth (e.g., thick basally, occlusally tapered and recurved), scapula (e.g., corpus slightly curved), sternum (e.g., lateral trabeculae laterally splayed with triangular distal expansions), and the foot (e.g., enlarged pedal digit II and long digit III claw). The unusual pedal morphology of BMNHC Ph 756 and other bohaiornithids hints at a unique ecological role played by this enantiornithine clade (Wang et al., 2014). Comparative measurements of the third toe have been shown to be indicative of ecology in living birds (Hopson, 2001; Zhou & Farlow, 2001). Adding the bohaiornithids to this data indicates that these birds were ecological generalists, intermediate between cursorial and arboreal birds (Wang et al., 2014). However, the hypertrophied third pedal claw, the robust digit II, and well-developed hallux may point to ecological specializations not found among other enantiornithines. BMNHC Ph 756 improves our understanding of the anatomy of Zhouornis hani, making this bohaiornithid the best known species of this newly recognized clade of enantiornithine birds. Institutional Abbreviations

BMNHC Beijing Natural History Museum, China

CNU Capital Normal University, Beijing, China

IVPP Institute of Vertebrate Paleontology and Paleoanthropology, Chinese Academy of Sciences, Beijing, China

LACM Natural History Museum of Los Angeles County, USA

We thank Stephanie Abramowicz for photography and figure preparation, Maureen Walsh for specimen preparation, and Lisa Granados for editorial assistance. We are grateful to J Thewissen, Wang Min, and an anonymous reviewer for their comments and suggestions to the original manuscript. J Siegel provided assistance with the X-ray imaging.

Additional Information and Declarations

Competing Interests

Author Contributions

Luis Chiappe is an Academic Editor for PeerJ.

Yuguang Zhang and Liu Di curation and infrastructure.

Jingmai O’Connor conceived and designed the experiments, performed the experiments, analyzed the data, wrote the paper, reviewed drafts of the paper.

Meng Qingjin reviewed drafts of the paper, curation and infrastructure.

Trond Sigurdsen performed the experiments, analyzed the data, wrote the paper, reviewed drafts of the paper.

Luis M. Chiappe conceived and designed the experiments, performed the experiments, analyzed the data, wrote the paper, prepared figures and/or tables, reviewed drafts of the paper.

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
