# Peer review of "New information on the anatomy of the Chinese Early Cretaceous Bohaiornithidae (Aves: Enantiornithes) from a subadult specimen of Zhouornis hani"

_PeerJ, doi:10.7717/peerj.407_

## Round 0.1 · original submission · Major Revisions

Whereas the reviewers consider this an important specimen that elucidates a number of anatomical features of this paleontologically significant species, they also identify problems. In my view, it is important that you clarify the collecting/provenance status of the specimen, especially in view of the large number of forgeries that affect avian Cretaceous paleontology. If you cannot provide provenance information, say so, and explain what you do know. I also believe that it is necessary for you to limit the discussion section by removing the speculative functional morphology section. Regarding the tentative statements regarding the age of this individual, make your assumptions more explicit.

Reviewer 1 ·

Basic reporting

Overall this is a good paper. The descriptions and illustrations are excellent. It is good to have another specimen of this taxon for comparison, and to fill in the anatomical gaps left by the holotype.

The major places where this manuscript is deficient relate to the non-anatomical details. First, there is no information about provenience or source of the specimen. Second, no phylogeny. Third, no control on statements about chronological age of specimens. Fourth, speculations about functional morphology are poorly constrained and not tested explicitly, and mostly should be eliminated.


Materials and methods

1. Usually a locality map is standard. It is not good enough for the authors to say “possibly the Jiufotang Formation.” The specimen is scientifically worthless without better locality information.

2. The authors also need to detail precisely how and from whom the specimen was acquired. This is because many of these specimens have faked parts and are composites of other specimens. The scientific community will not be able to understand and expose which agents are selling altered specimens, and where they are coming from, unless this information is published up front in each and every manuscript on these specimens – some of which should never have been published in the first place,.

3. It would also be good to have a phylogeny to show readers where this taxon is placed.


Discussion

327: You cannot say that a specimen is younger simply because it is smaller. If you have no independent evidence why it is younger, omit the statement, please. As we have learned, statements about relative size and relative fusions of bone are unreliable guides to maturity. Histological analysis on the femora or tibiae of this and the type specimens could address this problem; no other line of evidence is known to be able to do so.

333-338: We have learned time and again that relative size is not a good indicator of maturity. Neither is lack of fusion. If the specimens register an EFS in their outer cortex, they have effectively ceased linear growth. No other indicators are reliable. It is not clear on what basis Zhang et al. (2013) decided that the holotype was sexually mature, and how they used histology to do it; please explain.

345 and many places following: please do not use the term “ecological niche.” A niche is specific to a species, and it encompasses all aspects of the animal's life history strategy, adaptations, and ecological habits. Use the term “adaptive zone” or “resource zone” or “adaptation,” or “ecological role,” depending on what you mean.

354, 360, 366, etc.: “predatorial” is not an English word; please use “predatory”

366-370: this is a bit restrictive in thinking that you can have a particular adaptation or ecological role only if you have exactly the same morphology as a group of living birds has. If you want to discuss functional morphology at all – and you should not, unless you follow its principles – you have to identify the key features that diagnose the ability to perform one function or another. You have not done this here. Wang et al. (2014) were closer to the mark, and I suggest a deeper reading of their paper and an analysis that follows the principles that they describe rather than simply reacting to what they (and Zhang et al. 2013) wrote and going by general similarities. Do you really have evidence that these birds were hunting aerial insects with those feet? Please regard the principles and practices of functional morphology enough to avoid speculation.

In the final paragraph of the Discussion, you overlook a very important feature of these claws. You are only considering two-dimensional shape. How robust were they in three dimensions? If this is unknown, it is fruitless to speculate about functions. Claw curvature, length, and robustness are different ecological indicators.

This paper could actually do without the functional/ecological speculations because none of them are explicitly tested, and the authors already have a very good paper restricted to anatomy.

Experimental design

no comments

Validity of the findings

see above for areas in which the validity is questionable.

Additional comments

no further comments, except that the description is perfectly fine.

Reviewer 2 ·

Basic reporting

This manuscript was well excuted and I'm looking forward to seeing it published in Peer J. My detail comments are added to draft, and here are my general comments.

1. All the figures are well taken, but some labbeling in fig. 3 and 4 need to be replaced in order to specify the exact bone element;
2. Additional labeling on Fig. 5, especiialy the caudal vertebrae are welcomed. Besides, the incompelet fusion present in the caudal end of pygostyly is barely visible from fig.5. I would suggest that either a new photo of pygostly is taken or a line drawing is needed.
3. I would like to recommend that the author provide a photograph of whole skeleton of Zhouornis hani holotype, if it is possible.
4. The genus name is Longusunguis, rather than Longiungulavis, and corrected this throughtly.
5. The ecological adaption of Zhouornis is interesting, and any expanded discussion of this part is wlecomed. I have to admit that statistic approch to infer ecological niches of fossil birds is not straightforwad, although many researches have been conducted. My sussgestion is that whether the authors could provide photograps of foot of typical bird or prey (it would be best just the skeleton), which they think are comparable with Zhouornis. What about line drawing, if those modern birds are not available. This comparison would benefit readers a lot.

Experimental design

I'm very interested in the X-ray photos in this manuscript, and would like to see if the author could explicit parameters and working condition of this method in "materials and method" section.

Validity of the findings

I agree with the authors that referring the new species to Zhouornis and the general conclution about the ecological inferrence. All the disscussion are based on sound anatomical description and comparision.

Additional comments

The paper describes a new referred specimen of Zhouornis hani, and provided detail anotomical description and comparsion, which indeed help to advance our understading of morhology of Enantiornithes. Additionally, the authors proposed predatory specilization for Zhouornis based on general comparison with modern birds, hinting a novel niches for these dominating avian group. I recommened this manuscript to be published in Peer J with minor reivison.

Annotated reviews are not available for download in order to protect the identity of reviewers who chose to remain anonymous.

---

## Round 0.2 · accepted · Accept

Thank you for amending your manuscript to accommodate the suggestions by the reviewers and myself.